# SCA^®^ Slows the Decline of Functional Parameters Associated with Senescence in Skin Cells

**DOI:** 10.3390/ijms23126538

**Published:** 2022-06-10

**Authors:** Begoña Castro, Naiara de Paz, Salvador González, Azahara Rodríguez-Luna

**Affiliations:** 1Histocell S.L., Bizkaia Science and Technology Park, 48160 Derio, Spain; bcastro@histocell.com (B.C.); info@histocell.com (N.d.P.); 2Department of Medicine and Medical Specialties, Faculty of Medicine, Alcalá de Henares University, 28805 Madrid, Spain; salvagonrod@gmail.com; 3Cantabria Labs, Innovation and Development Department, 28043 Madrid, Spain; 4Department of Pharmacology, Faculty of Pharmacy, University of Seville, 41012 Seville, Spain

**Keywords:** skin aging, *Cryptomphalus aspersa* secretion (SCA^®^), cell senescence, mTOR, metabolic homeostasis

## Abstract

The identification of compounds and natural ingredients that can counteract tissue stress and dysfunction induced by aging in skin cells is warranted. Here, we investigated the activity of the secretion from the snail *Cryptomphalus aspersa* (SCA^®^), an active compound with well-established beneficial effects on skin integrity and aging. To determinate its senescence-regulation mechanisms, we used a model where damage was induced by hydrogen peroxide (H_2_O_2_). The results showed that SCA^®^ positively modulated factors involved in cell senescence such as β-galactosidase and cell morphology, secretory efficiency markers (SIRT1/6 and carboxymethyl-lysine), and metabolic and redox homeostasis (mTOR and ROS). This study demonstrated a novel compound that is activity-modulating, reduces cell senescence, and increases longevity to maintain skin homeostasis and functionality.

## 1. Introduction

Human skin is exposed to two major factors: intrinsic features, such as genetic variability and metabolic activity, and external stressors, which determine the “aged” phenotype [1]. The cumulative process of aging is remarkably complex and contextual, and stems from both direct damage to cell and tissue structures by external agents and reactive metabolic byproducts such as reactive oxygen species (ROS), and the progressive loss of homeostatic surveillance and tissue repair mechanisms counteracting such damage [2].

During aging progression, the skin’s ability to repair cell and tissue damage and reduce inflammation significantly declines, promoting a low-grade, pro-inflammatory state with imbalanced inflammation and oxidative stress markers, known as “inflammaging” [3,4,5]. When sustained, this altered inflammatory environment contributes to the progression of tissue damage and loss of metabolic homeostasis, triggering cell senescence, a term used to refer to irreversible cell cycle arrest, cell morphology changes, epigenetic alterations, and an aberrant secretory profile, among others [6,7,8].

DNA damage induced by oxidative stress, senescence-associated secretory phenotype (SASP) expression, and impaired autophagy are all general features of senescent cells. SASP is characterized by the upregulated secretion of several families of soluble and insoluble factors, such as interleukins, chemokines, growth factors, proteases, and extracellular matrix (ECM) insoluble factors [9]; the SASP secretome promotes certain tissue microenvironment features, such as the inhibition of cell proliferation and enhanced inflammation (presumably to facilitate senescent cell clearance) [6,10,11]. Recent data suggest that fibroblasts are the main factors determining skin aging, as a result of alterations to their proliferation levels and dermal accumulation of SASP fibroblasts [12]. Fibroblasts are key elements of dermal architecture and function, producing and arranging the components of the ECM, such as collagen [3], elastin, and GAGs; the loss of the ECM architecture further contributes to aging progression [3,12].

Hydrogen peroxide (H_2_O_2_) is a key source of DNA damage through ROS production in cells, which is closely associated with cellular senescence. Senescent cells commonly display increased cytosolic activity of β-galactosidase (β-gal), a senescence-associated biomarker originally discovered by Dimri et al. [13]. Furthermore, the increase in cyclin-dependent kinases and the presence of N6-carboxylmethyllysine (CML) arising from glycation reactions are senescent phenotypic traits [3]. In aged cells, the accumulation of advanced glycation end products (AGEs) stems from the reduction in sugars and other molecules, such as proteins or lipids, in the dermal ECM. Proteins such as collagen can be negatively affected by this phenomenon, reducing structural support and elasticity and thus further contributing to a senescence-like phenotype [14].

Increasing attention is also being placed on the potential role of melanocytes in contributing to skin aging processes. Melanocytes are a particular highly secretory cell population, where cumulative mutations can trigger both proliferation and senescent arrest [15], affecting proliferative potential and the differentiation of basal stratum keratinocytes [16]. Furthermore, melanocytes have recently emerged as relevant immune system modulators [17] and can contribute to SASP [18], classically attributed to hypodermal fibroblasts. Melanization is an evolutionarily early hallmark of persistent inflammation, and inflammatory dysregulation of the skin commonly entails altered melanization [19].

Among natural compounds and actives identified as having the potential to promote skin homeostasis and counteract the aging-associated decline in skin integrity, a promising source is a secretion from the snail *Cryptomphalus aspersa* (hereon, SCA^®^) [20]. SCA^®^ has shown significant activity in vitro and in vivo, both in protecting against damaging agents and in reducing aging-associated decline of the skin [20,21,22]. Specifically, the secretion composition features proteins of high and low molecular weight with fibroblast growth factor (FGF)-like activity, antioxidant molecules (superoxide dismutase (SOD), glutathion S transferase (GSH-T)), collagenase, hemocyanine, and other polypeptides, ECM building blocks (proteoglycans, glycosaminoglycans, and hyaluronic acid), and calcium salts. SCA^®^ exerts a positive impact on different processes involved in skin homeostasis maintenance and repair; fibroblast proteostasis, proliferation, and migration; ECM assembly; and effective wound healing [20,22,23,24,25]. However, in-depth characterization of the molecular mechanisms associated with these activities across different cell types is needed to establish ideal treatment protocols and possible synergies with other available compounds.

In this study, we described the impact of using SCA^®^ to treat both fibroblasts and B16-F10 melanoma cells that were induced to a senescent state through H_2_O_2_-mediated oxidative stress, by systematically profiling different biomarkers of their metabolic stress, proteostasis, and secretory capacity. We found that SCA^®^ attenuated senescence-associated parameters, such as ß-gal activity, and ROS levels in both fibroblasts and melanoma cells. These results correlated with a full reversal of expression levels of different indicators of secretory and metabolic homeostasis sensitive to senescent states, such as SIRT1/6, procollagen, mammalian target of rapamycin (mTOR) kinase, and CML accumulation, a marker of glycosylation, both an indicator of and a contributor to aging [26]. As will be discussed below, our observations suggest that SCA^®^ has the potential to help reduce tissue function decline during aging by preserving specific outputs in senescent cells, such as metabolism and lysosomal function, secretory efficiency, and proteostasis in both cell lines. These findings support the design of therapeutic/preventive strategies to effectively act on proliferative/regenerative skin niches.

## 2. Results

### 2.1. Impact of SCA on β-Galactosidase Activity

The effect of SCA^®^ treatment on general senescent phenotype of cultured human fibroblasts and mouse melanoma B16-F10 cells was analyzed. It has been reported that the exposure of various cell types to sub-lethal concentrations of H_2_O_2_ induces senescence [27,28,29]. Thus, cells were first exposed to 0.5 mM H_2_O_2_ for 2 h to induce oxidative stress-associated premature senescence. Cells were subsequently washed and treated with either fresh culture medium (control group) or SCA^®^ (at a concentration of 100 µg/mL for fibroblasts and 25 µg/mL in case of B16-F10 cells) for 96 h, at which time the percentage of β-galactosidase-positive cells was computed (Figure 1). Exposure to H_2_O_2_ significantly induced senescence in fibroblasts and melanocytes (*p* < 0.05). SCA^®^ treatment reduced the percentage of β-galactosidase-positive cells by 67% and 70% in fibroblasts and melanocytes cultures, respectively (** *p* < 0.01, * *p* < 0.05; Figure 1).

### 2.2. Impact of SCA^®^ on Intracellular ROS Levels

Intracellular ROS levels were determined by fluorescent 2′,7′-dichlorofluorescin diacetate (DCFH-DA) ROS-sensitive probe accumulation across the same conditions as indicated above. As expected, an increase in DCFH-DA fluorescence (equivalent to ROS accumulation) was observed in cells exposed to H_2_O_2_. On the contrary, cells treated with SCA^®^ experienced a significant decrease in DCFH-DA fluorescence compared with cells not treated with SCA (by 72.5% and 72.6%, *** *p* < 0.0001, in fibroblasts and B16-F10 cells, respectively (Figure 2)).

### 2.3. Impact of SCA on Senescent-Related Cell Morphology

To assess whether SCA^®^ treatment has an impact on senescence-related morphological changes, we examined a cytoskeleton with immunofluorescence after H_2_O_2_ and H_2_O_2_+SCA^®^ treatments. Senescent fibroblasts are characterized by a flattening and enlargement of the cell shape [30], which were observed in H_2_O_2_-exposed cells (Figure 3b). However, cells subsequently treated with SCA^®^ recovered basal cytoskeleton microtubule elongation and size compared with senescent cells (Figure 3c).

### 2.4. Analysis of Senescence-Sensitive Proteins

We then assessed the expression levels of different proteins sensitive to senescence induction by immunofluorescent staining upon exposure to H_2_O_2_ in the presence or absence of SCA^®^. One of the most representative senescence markers is mTOR, which is a crucial regulator of autophagy initiation and is involved in senescence through SASP regulation [31]. H_2_O_2_-induced senescence caused an increase in mTOR levels, while treatment with SCA^®^ reversed this effect and inhibited mTOR expression (Figure 4).

In the same context, sirtuins are longevity-related proteins that play an important role in DNA-damage repair and controlling inflammation and oxidative stress [32]. In this study, we evaluated the expressions of SIRT1 and SIRT6 after treatments and observed that H_2_O_2_ reduced SIRT1 and SIRT6 production. However, the subsequent incubation with SCA^®^ promoted the expressions of SIRT1 (Figure 5) and SIRT6 (Figure 6), reverting the suppression induced by H_2_O_2_ in senescent cells.

Advanced glycation end products (AGEs) also play important roles in skin aging. Different AGEs, such as carboxymethyl-lysine (CML) and pentosidine, cause glycation reactions; thus, AGE accumulation leads to a modification of skin homeostasis and promotes senescence phenotype [14]. To further investigate the impact of SCA^®^ treatment on senescent cell state, we analyzed the presence of CML after H_2_O_2_ and H_2_O_2_+SCA treatments by immunofluorescence. As images in Figure 7 show, H_2_O_2_ induced an increase in CML in fibroblasts and melanocytes, whereas a significant decrease in CML was observed in cells treated with SCA^®^ (Figure 7).

After exposure to H_2_O_2_, one of the most altered insoluble factors is collagen [33]. In this study, we evaluated the effect of H_2_O_2_ and H_2_O_2_+SCA^®^ treatments on the expression of procollagen-1. As observed in Figure 8, the expression of procollagen-1 was reduced in cells exposed to H_2_O_2_. However, in cells treated with SCA after being exposed to H_2_O_2_, the expression of this factor was restored to basal level (Figure 8).

## 3. Discussion

The identification of compounds modulating specific features of tissue and cell aging (senostatic/senomorphics) or clearing senescent cells to slow the progression of tissue homeostasis-decline (senolytics) is an intensive field of research [34]. Thus, characterization is warranted of the cellular and molecular mechanisms by which certain active compounds exert their beneficial action on the preservation and repair of skin structure and delay in the aging process [35].

In this study, we investigated the specific impact of SCA^®^ across phenotypic aspects associated with acute oxidative-stress-induced cell senescence. We found that SCA^®^ treatment recovered cell morphology and normalized indicators of metabolic homeostasis (ROS, SIRT1/6), secretion capacity (CML, procollagen), and proteostasis (as supported by secretory homeostasis, as well as potential molecular indicators of lysosomal integrity (mTOR and β-galactosidase levels)). These observations suggested that SCA^®^ can contribute to reduced skin aging progression by sustaining and improving the functionality of aged/senescent skin cell populations, reducing tissue integrity decline, and thus the degenerative feedback loops at play in aging tissue. These results both confirm and highlight activities and anti-aging benefits observed in recent studies of different *C. aspersa* secretion preparations, which have demonstrated powerful activity on fibroblast and keratinocyte proliferation and migration, increasing wound healing [36,37,38,39]. Thus, SCA^®^ was proposed as a regenerating ingredient, exhibiting a capability for rejuvenating the composition and functionality of damaged and aging skin [37].

Though senescent cells are characterized by altered cell-cycle arrest, our study demonstrated that the reversal of β-galactosidase accumulation does not necessarily correlate with cell cycle profiles indicative of blockade onto the G0 stage (data not shown). Thus, changes in these parameters should be carefully interpreted when assessing the impact of different treatments and inferring underlying molecular mechanisms. In this particular case, we interpret the reversal of β-galactosidase accumulation upon exposure to SCA^®^ as indicative of improved lysosomal integrity (in agreement with the mTOR levels) and proteostasis in senescent cells, which in turn prevents tissue functional decline. Furthermore, senescent cells accumulated by chronological skin aging reduced antioxidant capacity, promoting oxidative processes and altering cell structure, which contribute to age-related skin changes [38]. SCA^®^ considerably reduced H_2_O_2_-induced oxidative stress and reversed senescence phenotype.

mTOR signaling is a well-known marker of senescence through SASP regulation [39] and cell structural changes promotion [40], among other mechanisms. Thus, over-activation of mTOR promotes the induction of cellular senescence, while its repression blocks cellular senescence, reduces SASP, and prolongs cell life span [41]. Therefore, the autophagy regulator mTOR plays a central role in determining aging-associated processes [34]. Sirtuins are other regulators slowing cell and organism aging [42,43,44], although the precise mechanisms by which this is attained are unclear. Sirtuins can modulate cell metabolic state and proteostasis at different levels [42,45,46,47,48], as well as counteract genome reprogramming [49,50]. Sirtuins were proven to exert an anti-aging effect on skin components, reducing inflammaging and promoting autophagy [51]. The treatment of senescent fibroblasts with SCA^®^ repressed mTOR and promoted SIRT1/6 expression, reducing the senescence process. It would be interesting to further explore the specific contribution of sirtuins to the biological effects of SCA^®^.

Advanced glycation end products (AGEs) can induce mTOR expression and cause morphological changes characteristic of senescence phenotypes. AGE accumulation disrupts the balance between the synthesis of ECM components and enzymes, altering skin cell biology and thus skin structure. Furthermore, these sub-products can increase intracellular ROS and pro-inflammatory cytokines, altering secretory activity [14]. Treatment with SCA^®^ reduced ROS production and the accumulation of AGEs such as CML, which up-regulates glycation reactions. With respect to collagen production, Huang et al. demonstrated that oxidative stress induced by low H_2_O_2_ concentrations exposure can trigger skin repair processes, thereby increasing type I procollagen expression [52]. Nevertheless, Park et al. showed that H_2_O_2_ reduces the mRNA expression of type I collagen [53], and disrupts transforming growth factor beta transduction and subsequently inhibits collagen biosynthesis, promoting skin aging [54]. We demonstrated that SCA^®^ treatment increases procollagen-1 secretion in senescent cells, reducing the aged phenotype provoked by H_2_O_2_. Importantly, the deposition and maintenance of ECM, serviced by competent cells, is an input as essential as circulating growth factors to ensure tissue repair and health [55,56,57,58,59], and likely contributes to the curbing of tissue aging.

We also sought to investigate the specific impact of SCA^®^ on melanocytic lineages. Melanocytes have emerged as central elements in the complex cell-communication networks responsible for organizing skin architecture and function, but our understanding as to how these cell populations experience aging and contribute to tissue decline is very limited. Altered melanization is a hallmark of aging. Our observations clearly demonstrated that SCA promotes proteostasis and reduces oxidative stress in B16-F10 melanocytes. In line with the results obtained in fibroblasts, SCA^®^ does not affect cell cycle stopping in B16-F10, but does inhibit mTOR and promote SIRT1/6 expression. SCA^®^ also reduced cell senescence in B16-F10 cells through glycation product modulation. These findings demonstrated the role of SCA^®^ in senescence modulation not only in fibroblasts, but also in a melanocytic cell linage. The fact that SCA^®^ can inhibit the senescence of melanoma cell lines suggests a further anti-aging mechanism, including the normalization of melanogenesis. Further studies into the role of normal human-epidermal melanocytes in skin aging processes would help shed further light on these implications.

Cellular senescence is the object of intensive research for the development of effective anti-aging strategies. Though molecular crosstalk among the senescence mechanisms is unclear, senescence reversal is a promising treatment to promote skin well-aging [60]. Our results position SCA^®^ as an active compound contributing to skin repair and health and inhibiting aging by maintaining autophagy functions and tissue integrity, even in the face of cell senescence. These insights are valuable to better understand SCA’s action mechanisms, and to determine its role in anti-aging treatments and potential synergies with other compounds in order to promote, for example, the proliferation and renewal of skin stem cell populations. Further considerations and more detailed forthcoming experimental approaches will be necessary to complete the understanding of SCA^®^ activity against senescence.

## 4. Materials and Methods

### 4.1. Cell Culture

Primary human dermal fibroblasts (HDFs) were isolated from human skin kindly donated by patients that had undergone aesthetic surgery after signing an informed consent form previously approved by the corresponding ethical committee (Ethics Committee for Drug Research of Euskadi (Basque Country, Spain)) with the approval code E09-28. Fibroblasts were obtained as previously described by Wang et al. and cultured in DMEM (Sigma–Aldrich D5796, St. Louis, (MO), USA) plus FBS 10%, 100 U/mL penicillinm and 100 mg/mL streptomycin (HyClone Laboratories, South Logan, (UT), USA), under the same incubation conditions [61]. The B16-F10 mouse melanoma cell line was kindly provided by Dr. Ana Alonso-Varona from the Department of Cell Biology and Histology, Faculty of Medicine, University of the Basque Country (Leioa-Bizkaia, Spain). Both the HDFs and melanocytes B16-F10 were cultured in Dulbecco’s modified Eagle medium (DMEM) supplemented with 10% (*v*/*v*) fetal bovine serum (FBS). Cells were maintained at 37 °C, 5% humidity, and 5% CO_2_ in an incubator (Heracell CO_2_ incubator 150i, 41077201, Thermo Fisher Scientific Inc., Waltham, (MA), USA).

### 4.2. Natural SCA^®^ Extract and Cell Treatment

A standardized SCA^®^ preparation was obtained and used as described by Cantabria Labs, (Madrid, Spain) [62]. Cell cultures were supplemented with 100 µg/mL (fibroblasts) or 25 µg/mL (B16–F10) of SCA^®^ and maintained after cell determinations at different times [36,37,39]. To evaluate the effects of SCA^®^ on senescent cells, fibroblasts and melanocytes were first exposed to H_2_O_2_ (as described in the next section) to induce senescence, and then supplemented with SCA^®^ by adding this compound to the cell medium.

### 4.3. Senescence Induction

Fibroblasts and melanocytes were seeded in 24-well culture plates at a density of 10.000 cells per well, and grown overnight in DMEM supplemented with 10% of FBS. To induce senescence, cells were exposed to 0.5 mM of H_2_O_2_ in DMEM without FBS for 2 h. After that, the culture medium was replaced, and fresh DMEM supplemented with 10% of FBS with or without SCA^®^ was added until measurements were performed.

### 4.4. Intracellular ROS Determination

Intracellular ROS levels were quantified using 2′,7′-dichlorodihydrofluorescein diacetate (DCFH-DA; Molecular Probes, Oregon, (AZ), USA). Fibroblasts were seeded in 96-well tissue culture plates at a density of 4000 cells/well. After senescence induction as previously described, the culture medium was replaced with or without SCA^®^ treatment and incubated for 18 h. After this period, the medium was removed and a fresh medium with 10 μM DCFH-DA was added. Plates were incubated at 37 °C for 30 min in the dark. The medium was then replaced with oxidizing media in the absence or presence of antioxidants. Fluorescence in the wells was measured every 5 min during 90 min (λex 485 nm/λem 528 nm; Thermo Fisher Scientific Inc., Waltham, (MA), USA).

### 4.5. β-Galactosidase Staining

The positive blue staining of β-galactosidase was used to estimate cellular senescence. For senescence determination, cells were seeded in 24-well plates at a concentration of 10,000 cells per well. β-galactosidase staining was analyzed at 96 h after senescence induction by 0.5 mM of H_2_O_2_, using a Senescence Cells Histochemical Staining Kit (Sigma–Aldrich CS0030, St. Louis, (MO), USA)). Briefly, the culture medium was aspirated from the wells, and the cells were washed twice with 1 mL of PBS. Then, 1.5 mL of fixing buffer was added per well and incubated for 7 min at room temperature. After washing twice with PBS, 1 mL of freshly prepared dying solution, prepared following kit instructions, was added to the wells, and incubated overnight at 37 °C without CO_2_. The plates were sealed with parafilm to avoid drying. Finally, the wells were washed with PBS, and the cells were scored for β-galactosidase-positive staining using light microscopy. Five representative images from each well were captured, total cell number and β-galactosidase-positive cells were obtained by using Image-J software (Oracle Corporation, Santa Clara, (CA), USA), and the percentage of positive cells with respect to total cells in each imaged was determined.

### 4.6. Immunofluorescence Analysis

For the immunostaining procedure, after senescence induction and incubation with SCA^®^ for 48 h, the wells were washed with PBS and incubated with 0.5 mL of paraformaldehyde (4%) in PBS for 15 min at room temperature. After washing twice with PBS, a Triton X-100 (0.1%) solution with 2% of BSA in PBS was used to permeabilize the cells for 10 min at 4 °C, and 25 mL of BSA (1%) in PBS for 1 h at room temperature was used to block unspecific binding. Primary antibodies anti-a-tubulin (ab195887, 1:250, Abcam, Cambridge, UK), anti-SIRT1 (19A7AB4, 0,5 μg/mL, Abcam, Cambridge, UK), anti- SIRT6 (MA5-24768, 1:100, Thermo Fisher Scientific Inc., Waltham, MA, USA), anti-mTOR (AHO1232, 1:500, Thermo Fisher Scientific Inc., Waltham, MA, USA), anti-CML, N6-carboxylmethyl-lysine (ab125145, 1:200, Abcam, Cambridge, UK), and anti-procollagen type I (PCIDG10, 1:200, Sigma-Aldrich, St. Louis, MO, USA) (diluted with 1% BSA) were used to probe the tissues at 4 °C overnight. Subsequently, the secondary red fluorescence antibody (Alexa Fluor 568, 1:200, Thermo Fisher Scientific Inc., Waltham, MA, USA) was used to reveal the specific binding of the primary antibody. Finally, a nuclear counterstaining was performed with Hoechst solution (Hoechst 33258, 1:2000, H1398, Invitrogen by Thermo Fisher Scientific Inc., Waltham, MA, USA) for 20 min at room temperature.

### 4.7. Statistical Analysis

Data are represented as mean ± standard error of the mean (SEM) from at least three independent experiments. Quantitative determinations (β-galactosidase staining and intracellular ROS determination) were compared by Student’s t-test, using GraphPad Prism 5.00 (GraphPad Software, Inc., San Diego, CA, USA). Differences were interpreted as significant when *p* ≤ 0.05.

## Figures and Tables

**Figure 1 ijms-23-06538-f001:**
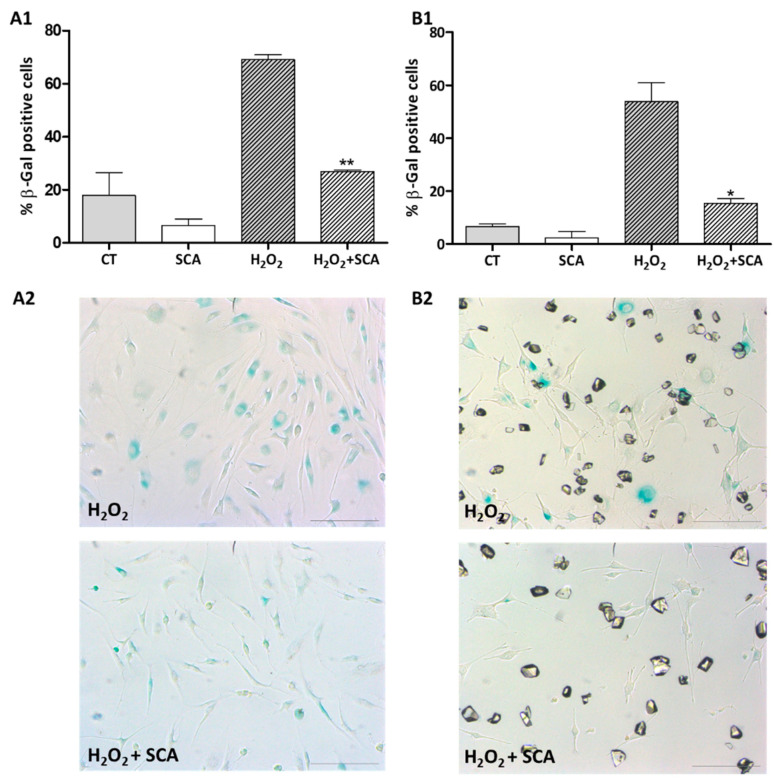
Determination of senescence, β-galactosidase staining. (**A1**,**B1**) Percentage of β-galactosidase-positive cells at different conditions (H_2_O_2_ vs. H_2_O_2_ + SCA^®^ ** *p* < 0.005, * *p* < 0.05; H_2_O_2_ vs. C * *p* < 0.05). (**A2**,**B2**) Representative images of β-galactosidase blue staining. A column corresponds to fibroblasts, and B column corresponds to B16-F10 cells. (C: Control Cells; SCA^®^: control cells treated with SCA^®^, H_2_O_2_: Aged control cells, H_2_O_2_ + SCA^®^: aged cells treated with SCA^®^). Scale bars: 200 μm.

**Figure 2 ijms-23-06538-f002:**
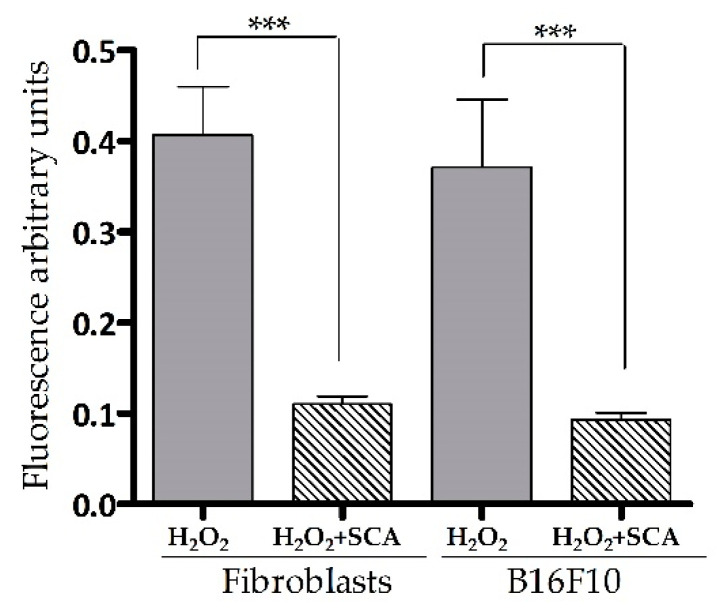
SCA^®^ treatment reduced ROS accumulation in senescent cells exposed to H_2_O_2_. Intracellular ROS levels. Arbitrary fluorescence units of DCFH-DA probe showing the presence of intracellular ROS in senescent fibroblasts and B16-F10 cells (H_2_O_2_) and senescent fibroblasts and B16 cells treated with SCA^®^ (H_2_O_2_ + SCA) (*** *p* < 0.0001).

**Figure 3 ijms-23-06538-f003:**
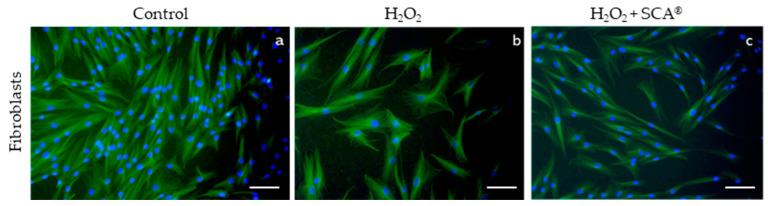
SCA^®^ treatment reversed morphological changes associated with senescence. Representative tubulin cytoskeleton images of control fibroblasts (**a**), fibroblasts treated with H_2_O_2_ (**b**), and fibroblasts treated with H_2_O_2_ + SCA (**c**). Representative inverted fluorescence microscopy images immunostained with anti-a-tubulin antibody appear in green, while nuclei that were stained with Hoechst probe appear in blue. Scale bars: 200 μm.

**Figure 4 ijms-23-06538-f004:**
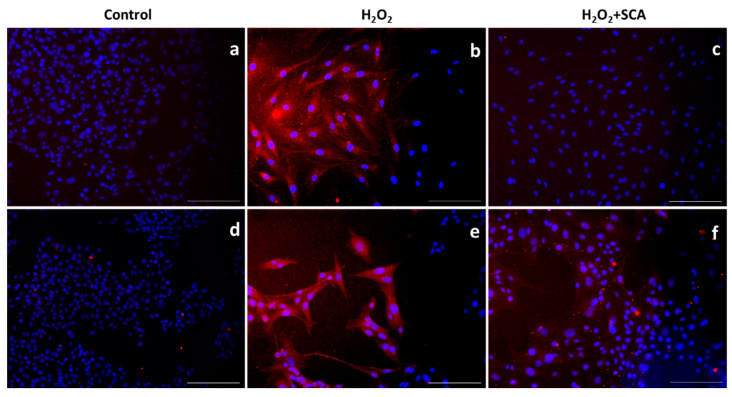
SCA^®^ treatment markedly inhibited mTOR activation triggered by H_2_O_2_ exposure. Immunostaining of mTOR. Representative inverted fluorescence microscopy images immunostained with anti-mTOR antibody appear in red, while nuclei stained with Hoechst probe appear in blue. Fibroblasts (**a**–**c**) and B16-F10 cells (**d**–**f**) exposed to H_2_O_2_ (**b**,**e**) and H_2_O_2_-exposed treated with SCA^®^ (**c**,**f**). Scale bars: 200 μm.

**Figure 5 ijms-23-06538-f005:**
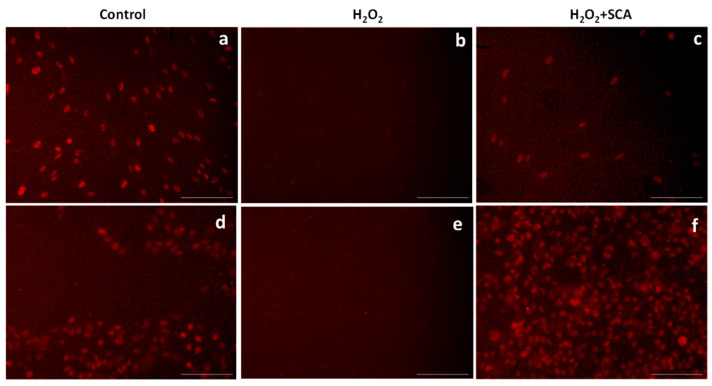
Treatment with SCA^®^ markedly induced SIRT1 expression in senescent cells. Representative fluorescence microscopy images after SIRT1 (red) immunostaining. Cells treated with H_2_O_2_ (**b**,**e**) showed decreased expression of SIRT1. Fibroblasts (**a**–**c**) and B16-F10 cells (**d**–**f**) were exposed to H_2_O_2_ and treated with SCA^®^ (**c**,**f**). Scale bars: 200 μm.

**Figure 6 ijms-23-06538-f006:**
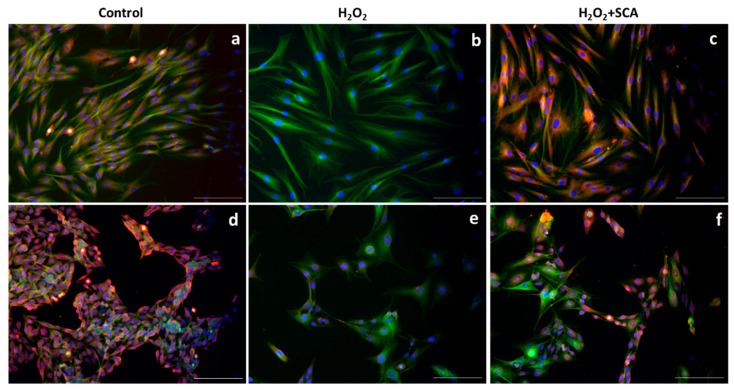
Treatment with SCA^®^ significantly promoted SIRT6 expression in senescent cells. Representative fluorescence microscopy images after SIRT6 (red) immunostaining. Green color shows cytoskeleton staining with anti-a-tubulin, and nuclei were stained blue with Hoechst probe. Fibroblasts (**a**–**c**) and B16-F10 cells (**d**–**f**) exposed to H_2_O_2_ (**b**,**e**) and H_2_O_2_-exposed cells and treated with SCA^®^ (**c**,**f**). Scale bars: 200 μm.

**Figure 7 ijms-23-06538-f007:**
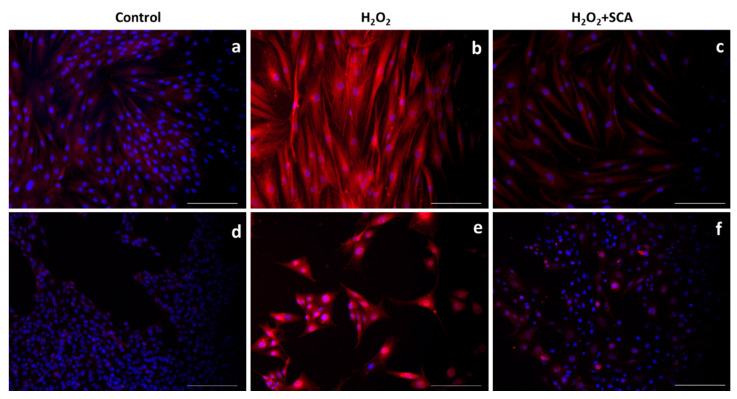
SCA^®^ significantly reduced CML immunostaining in H_2_O_2_-exposed cells. Representative inverted fluorescence microscopy images immunostained with anti-CML antibody in red. Nuclei were stained with Hoechst probe in blue. Fibroblasts (**a**–**c**) and B16-F10 cells (**d**–**f**) exposed to H_2_O_2_ (**b**,**e**) and H_2_O_2_-exposed cells and treated with SCA^®^ (**c**,**f**). Scale bars: 200 μm.

**Figure 8 ijms-23-06538-f008:**
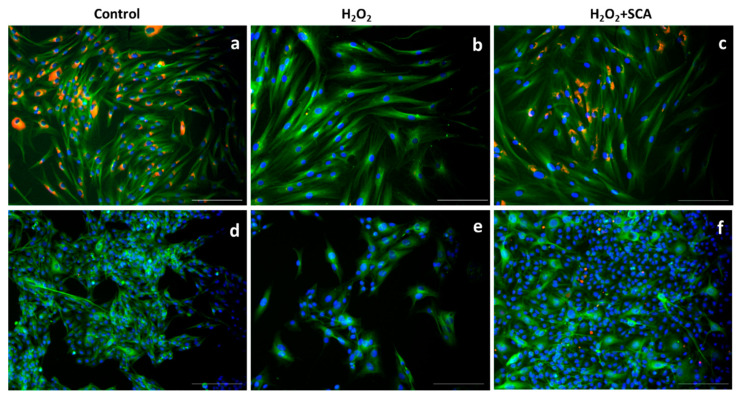
SCA^®^ treatment promoted procollagen-1 production. Representative inverted fluorescence microscopy images immunostained with anti-procollagen antibody in red, localized mainly at perinuclear level. Green color indicates cytoskeleton staining with anti-a-tubulin, and nuclei were stained blue with Hoechst probe. Fibroblasts (**a**–**c**) and B16–F10 cells (**d**–**f**) exposed to H_2_O_2_ (**b**,**e**) and H_2_O_2_-exposed cells and treated with SCA^®^ (**c**,**f**). Scale bars: 200 μm.

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
