# Peer review of "SCA^®^ Slows the Decline of Functional Parameters Associated with Senescence in Skin Cells"

_ijms, 2022, doi:10.3390/ijms23126538_

Round 1

Reviewer 1 Report

While the submitted manuscript suggests a potential benefits of secretions of the snail Cryptomphalus aspersa (SCA), there are many points of revisions before being considered for publication. 

First of all, reviewer think the title of manuscript is not properly written. While the measured parameters in the submitted study were associated with senescent cells, they are not solely associated with senescence at all. On the contrary, most of the parameters are actually linked with the cellular oxidative stresses. While the chronic oxidative damages can induce the cellular senescence, there are limited experimental evidences of cellular senescence in the submitted manuscript. The expressions of senescence marker proteins, at least p16 and cdk, should be provided. Also, using melanoma cells for investigating the cellular senescence can be a matter of debate. Reviewer strongly recommends to use normal human epidermal melanocytes instead. 

Secondly, there are many missing information in the results section. There are no scale bars or information of magnification in all the microscopic pictures. And what was the tested concentration of hydrogen peroxide, 0.5mM (from the results section) or 500mM (from the experimental section)? Also, how did authors determine the test concentration of both hydrogen peroxide and secretions of the snail Cryptomphalus aspersa? Are there any preliminary data about the potential cytotoxicity (in case of hydrogen peroxide) and effects on cellular proliferation (in case of secretions of the snail Cryptomphalus aspersa)? 

Author Response

We appreciate the comment made by the reviewer. However, we would like to highlight that treatment of different cell types with H2O2 is a common model for inducing stress-induced premature senescence (SIPS), and is one of the main targets for SCA compound. In this sense, in addition to demonstrating the oxidative induction by H2O2 (intracellular ROS levels), other parameters related to the senescence process has been analyzed. The typical morphological changes associated with senescent cells are shown, the increased expression of mTOR (a protein whose augmented expression alters the autophagic capacity of senescent cells, producing cell damage) and the decreased levels of SIRT1 and SIRT6. These proteins are essential for cell metabolism, and their decreased levels are also related to senescence. The decrease in SIRT proteins produce p53 acetylation, inducing the acquisition of a senescent phenotype. Furthermore, we observed an increase in cytosolic activity of β-galactosidase (β-gal), a biomarker associated with senescence (Dimri et al. 1995). Senescence also causes significant changes in cell morphology and the accumulation of advanced glycation end products (AGEs) observed in cells exposed to H2O2, highly linked to the senescence phenotype.

Dimri, Goberdhan P., Xinhua Lee, George Basile, Meileen Acosta, Glynis Scott, Calvin Roskelley, Estela E. Medrano, et al. 1995. “A Biomarker That Identifies Senescent Human Cells in Culture and in Aging Skin in Vivo.” Proceedings of the National Academy of Sciences of the United States of America 92 (20): 9363. https://doi.org/10.1073/PNAS.92.20.9363.

Also, using melanoma cells for investigating the cellular senescence can be a matter of debate. Reviewer strongly recommends to use normal human epidermal melanocytes instead. 

We appreciate and agree with the comment made by the reviewer. Nevertheless, several publications (cited below) have used this cell line in anti-aging studies since its use represents a good approximation and facilitates the experimentation (due to its culture and growth conditions) in contrast to the use of non-malignant melanocytes. Moreover, the use of malignant cells in this study does not interfere with the parameters that have been evaluated in the present work. Thus, the fact that SCA® can inhibit senescence of melanoma cell line would suggest a further anti-aging mechanism, including normalization of melanogenesis while oxidative and cell cycle levels are not altered. The relationship between melanogenesis and senescence modulation has previously been studied in b16 cells (Nanni et al., 2018, Cunha et al., 2012). However, we understand this point and included this limitation clearly in the manuscript.

Further studies into the role of normal human epidermal melanocytes in skin aging processes will of course help shed further light on these implications (line 267-268).

Cunha ES, Kawahara R, Kadowaki MK, Amstalden HG, Noleto GR, Cadena SM, Winnischofer SM, Martinez GR. Melanogenesis stimulation in B16-F10 melanoma cells induces cell cycle alterations, increased ROS levels and a differential expression of proteins as revealed by proteomic analysis. Exp Cell Res. 2012 Sep 10;318(15):1913-25. doi: 10.1016/j.yexcr.2012.05.019. Epub 2012 Jun 2. PMID: 22668500.

Nanni V, Canuti L, Gismondi A, Canini A. Hydroalcoholic extract of Spartium junceum L. flowers inhibits growth and melanogenesis in B16-F10 cells by inducing senescence. Phytomedicine. 2018 Jul 15;46:1-10. doi: 10.1016/j.phymed.2018.06.008. Epub 2018 Jun 15. PMID: 30097108.

Gam, D. H., Hong, J. W., Kim, J. H., & Kim, J. W. (2021). Skin-Whitening and Anti-Wrinkle Effects of Bioactive Compounds Isolated from Peanut Shell Using Ultrasound-Assisted Extraction. Molecules (Basel, Switzerland), 26(5), 1231. https://doi.org/10.3390/molecules26051231

Lorrio, S., Rodriguez-Luna, A., Delgado-Wicke, P., Mascaraque, M., Gallego, M., Perez-Davo, A., Gonzalez, S., & Juarranz, A. (2020). Protective Effect of the Aqueous Extract of Deschampsia antarctica (EDAFENCER) on Skin Cells against Blue Light Emitted from Digital Devices. International journal of molecular sciences, 21(3), 988. Doi: 10.3390/ijms21030988

Portillo, M., Mataix, M., Alonso-Juarranz, M., Lorrio, S., Villalba, M., Rodriguez-Luna, A., & Gonzalez, S. (2021). The Aque-ous Extract of Polypodium leucotomos (FernblockR) Regulates Opsin 3 and Prevents Photooxidation of Melanin Precursors on Skin Cells Exposed to Blue Light Emitted from Digital Devices. Antioxidants (Basel, Switzerland), 10(3), 400. doi: 10.3390/antiox10030400

Secondly, there are many missing information in the results section. There are no scale bars or information of magnification in all the microscopic pictures.

We fully agree with this appreciation from reviewer, revised microscopy images now include scale bars.  

And what was the tested concentration of hydrogen peroxide, 0.5mM (from the results section) or 500mM (from the experimental section)? Also, how did authors determine the test concentration of both hydrogen peroxide and secretions of the snail Cryptomphalus aspersa?

We apologize the mistake, the correct tested concentration is 0.5 mM. In the current version, we modified the experimental section (line 303). To determine H2O2 concentration, we considered the generation of a situation where senescence was clearly induced but the cells could have the ability to reverse the senescence process after SCA treatment. We performed a H2O2 concentration and treatment time sweep and we selected 0,5 mM dose for two hours for senescence induction on around 50% of cell population (Pieńkowska et al. 2020)(Kim et al. 2021)(Dimozi et al. 2015). Less concentration of H2O2 did not produce sufficient cell damage, so the cells could spontaneously recover soon after peroxide insult and higher concentration reach near 100% of senescent or so damaged cells, that did not allow to study senescence process.

Kim, Hyun Ji, Boram Kim, Hyung Jung Byun, Lu Yu, Tuan Minh Nguyen, Thi Ha Nguyen, Phuong Anh Do, et al. 2021. “Resolvin D1 Suppresses H2 O2-Induced Senescence in Fibroblasts by Inducing Autophagy through the Mir-1299/Arg2/Arl1 Axis.” Antioxidants 10 (12). https://doi.org/10.3390/antiox10121924.

Pieńkowska, Natalia, Grzegorz Bartosz, Monika Pichla, Michalina Grzesik-Pietrasiewicz, Martyna Gruchala, and Izabela Sadowska-Bartosz. 2020. “Effect of Antioxidants on the H2O2-Induced Premature Senescence of Human Fibroblasts.” Aging 12 (2): 1910–27. https://doi.org/10.18632/aging.102730.

Dimozi, A., E. Mavrogonatou, A. Sklirou, and Dimitris Kletsas. 2015. “Oxidative Stress Inhibits the Proliferation, Induces Premature Senescence and Promotes a Catabolic Phenotype in Human Nucleus Pulposus Intervertebral Disc Cells.” European Cells and Materials 30: 89–103. https://doi.org/10.22203/eCM.v030a07.

Are there any preliminary data about the potential cytotoxicity (in case of hydrogen peroxide) and effects on cellular proliferation (in case of secretions of the snail Cryptomphalus aspersa)? 

We thank the reviewer for his/her comment, because we think that we had a problem with symbology in the article. The concentrations were 100 ug/mL instead of mg/mL and we think that the same thing has happened with H2O2 concentration mistakes (500 ug/mL instead of 500 mg/mL). All concentrations have now been reviewed and corrected in the new version.

Nevertheless, we would like to clarify the selected SCA® concentration and we included some previous bibliography in the new version (line 297).

Reviewer 2 Report

The authors present novel data on cell senescence of a known compound. The manuscript is generally well written and easy to follow.

Please find my questions below:

 line 109, Figure 1: It is not clear to me, whether the images are w/ or w/o H2O2 and SCA. Please specify this by labelling the images properly.

line 134, Figure 3: I think scale bars would help in these images because H2O2 yields a senescent phenotype in which the cells are enlarged compared to control and treatment. This is also valid for figure 4.

line 159, Figure 5: Unfortunately, it looks like the background settings were not the same in all the images and this makes it difficult to judge the result. It looks as if b and e have less background than the other images. Can you correct for this?

line 267: ...skin aging processes with will of course...

line 284: please give the name of the ethical committee.

line 463,:reference is incomplete

line 532: citation 67 is already citation 37.

General remarks/questions:

-Why did the authors choose a mouse melanoma cell line, and what is the relevance for humans?

-I guess the staining of the nuclei was done with Hoechst staining, not with Hoescht staining. The authors have to adjust this throughout the manuscript.

Author Response

The authors present novel data on cell senescence of a known compound. The manuscript is generally well written and easy to follow.

We sincerely thank the positive appreciation and constructive comments made by the reviewer.

Please find my questions below:

 line 109, Figure 1: It is not clear to me, whether the images are w/ or w/o H2O2 and SCA. Please specify this by labelling the images properly.

We thank the reviewer for his/her comment, we have labelled the images to avoid misunderstanding.

line 134, Figure 3: I think scale bars would help in these images because H2O2 yields a senescent phenotype in which the cells are enlarged compared to control and treatment. This is also valid for figure 4.

We fully agree with this appreciation from reviewer, we revised microscopy images that now include scale bars. 

line 159, Figure 5: Unfortunately, it looks like the background settings were not the same in all the images and this makes it difficult to judge the result. It looks as if b and e have less background than the other images. Can you correct for this?

We thank the appreciation and we included new images. Also, figure legend was modified in the revised version.

Figure 5. Treatment with SCA® markedly induces SIRT1 expression in senescent cells. Representative fluorescence microscopy images after SIRT1 (red) immunostaining. Cells treated with H2O2 showed decreased expression of SIRT1. Fibroblasts (top panel a-c) and B16-F10 cells (bottom panel d-f) exposed to H2O2 and treated with SCA®. Scale bars: 200 μm

line 267: ...skin aging processes with will of course...

Thank you for citing the mistake, we have modified the new version.

line 284: please give the name of the ethical committee.

We appreciate the comment. We include “Ethics Committee for Drug Research of Euskadi (Basque Country, Spain)” (CEIm-E) at lines 284-285.

line 463,:reference is incomplete

Thank you for citing the mistake, we have modified the new version.

line 532: citation 67 is already citation 37.

 Thank you for citing the mistake, we have modified the new version.

General remarks/questions:

-Why did the authors choose a mouse melanoma cell line, and what is the relevance for humans?

We appreciate and agree with the comment made by the reviewer. Nevertheless, several publications (cited below) have used this cell line in anti-aging studies since their use represents a good approximation and facilitates the experimentation (due to its culture and growth conditions) in contrast with using non-malignant melanocytes. Moreover, the use of malignant cells in this study does not interfere with the parameters that have been evaluated in the present work. Thus, the fact that SCA® can inhibit senescence of melanoma cell line would suggest a further anti-aging mechanism, including normalization of melanogenesis while oxidative and cell cycle levels does not been disturbed. The relationship between the melanogenesis and senescence modulation has previously been studied in b16 cells (Nanni et al., 2018, Cunha et al., 2012). Nevertheless, we understand this point and we included this limitation clearly in the manuscript.

Further studies into the role of normal human epidermal melanocytes in skin aging processes will of course help shed further light on these implications (line 267-268).

Cunha ES, Kawahara R, Kadowaki MK, Amstalden HG, Noleto GR, Cadena SM, Winnischofer SM, Martinez GR. Melanogenesis stimulation in B16-F10 melanoma cells induces cell cycle alterations, increased ROS levels and a differential expression of proteins as revealed by proteomic analysis. Exp Cell Res. 2012 Sep 10;318(15):1913-25. doi: 10.1016/j.yexcr.2012.05.019. Epub 2012 Jun 2. PMID: 22668500.

Nanni V, Canuti L, Gismondi A, Canini A. Hydroalcoholic extract of Spartium junceum L. flowers inhibits growth and melanogenesis in B16-F10 cells by inducing senescence. Phytomedicine. 2018 Jul 15;46:1-10. doi: 10.1016/j.phymed.2018.06.008. Epub 2018 Jun 15. PMID: 30097108.

Gam, D. H., Hong, J. W., Kim, J. H., & Kim, J. W. (2021). Skin-Whitening and Anti-Wrinkle Effects of Bioactive Compounds Isolated from Peanut Shell Using Ultrasound-Assisted Extraction. Molecules (Basel, Switzerland), 26(5), 1231. https://doi.org/10.3390/molecules26051231

Lorrio, S., Rodriguez-Luna, A., Delgado-Wicke, P., Mascaraque, M., Gallego, M., Perez-Davo, A., Gonzalez, S., & Juarranz, A. (2020). Protective Effect of the Aqueous Extract of Deschampsia antarctica (EDAFENCER) on Skin Cells against Blue Light Emitted from Digital Devices. International journal of molecular sciences, 21(3), 988. Doi: 10.3390/ijms21030988

Portillo, M., Mataix, M., Alonso-Juarranz, M., Lorrio, S., Villalba, M., Rodriguez-Luna, A., & Gonzalez, S. (2021). The Aque-ous Extract of Polypodium leucotomos (FernblockR) Regulates Opsin 3 and Prevents Photooxidation of Melanin Precursors on Skin Cells Exposed to Blue Light Emitted from Digital Devices. Antioxidants (Basel, Switzerland), 10(3), 400. doi: 10.3390/antiox10030400

-I guess the staining of the nuclei was done with Hoechst staining, not with Hoescht staining. The authors have to adjust this throughout the manuscript.

Thank you for citing the mistake, we have corrected the new version.
